# Biodegradable Polyester Materials Containing Gallates

**DOI:** 10.3390/polym12030677

**Published:** 2020-03-18

**Authors:** Malgorzata Latos-Brozio, Anna Masek

**Affiliations:** Institute of Polymer and Dye Technology, Lodz University of Technology, 90-924 Lodz, Poland; anna.masek@p.lodz.pl

**Keywords:** polylactide (PLA), polyhydroxyalkanoate (PHA), gallates, antioxidant activity, stabilizers, packaging materials

## Abstract

Gallates are widely used as antioxidants in the food and cosmetics industries. The purpose of the study was to obtain pro-ecological materials based on biodegradable polyesters, such as polylactide (PLA) and polyhydroxyalkanoate (PHA), and gallates. Gallates (ethyl, propyl, octyl, and lauryl) have not been used so far in biodegradable polymers as stabilizers and indicators of aging. This manuscript examines the properties of gallates such as antioxidant capacity and thermal stability. This paper also presents the following analyses of polymer materials: specific migration of gallates from polymers, SEM microscopy, differential scanning calorimetry (DSC), wide-angle X-ray diffraction, mechanical properties, surface free energy, and determination of change of color after controlled UV exposure, thermooxidation, and weathering. All gallates showed strong antioxidant properties and good thermal stability. Due to these properties, in particular their high oxidation temperature, gallates can be successfully used as polyester stabilizers. Biodegradable polyesters containing gallates can be an environmentally friendly alternative to petrochemical packaging materials.

## 1. Introduction

Plastic waste is a serious environmental problem on a global scale. Packaging materials for food products constitute a significant part of plastic waste [1,2]. Currently, efforts are made to reduce the amount of this waste by replacing petrochemical polymers by easily degradable polymer materials obtained from biorenewable sources (such as alginate, cellulose, chitosan, collagen carrageenan, corn zein, soy, starches). Sustainable packaging materials are often more functional and environmentally friendly then traditional plastic [3,4].

Particularly noteworthy are biodegradable aliphatic polyesters, which are increasingly used as packaging materials. The most widely used biodegradable polymer is polylactide (PLA). This polyester can be derived from 100% renewable resources. Due to its moderately low production costs and good compliance with environmental requirements, PLA is widely used in many areas, including medicine (e.g., sutures, resorbable implants), packaging, and fibers [5,6,7]. Other important materials of the group of biodegradable polyesters are polyhydroxyalkanoates (PHAs) of microbiological origin and their synthetic analogues. Natural polyhydroxyalkanoate is produced in vivo by bacteria as well as by transgenic microorganisms and plants. Plant cells can produce only a small quantity of these polymers, whereas in bacteria, PHAs are accumulated at a level up to 90% of the dry cell mass. Synthetic analogues of PHAs offer greater advantages than natural-origin polymers because they can be tailored to impart a wider range of properties and to obtain tailor-made biodegradable materials for specific applications in different fields. Similar to PLA, PHAs are used in medicine, surgery, pharmacology, agriculture, and the packaging industry [8,9,10,11]. 

Increasing attention is being paid to the fact that, besides polymer matrices, also processing additives should be environmentally and consumer-friendly, especially those for food packaging and medical applications materials. Plant-based substances are more and more commonly proposed as pro-ecological stabilizers for polymeric materials, dyes, antibacterial/antifungal agents, freshness indicators of packaged food products, as well as indicators of the life time of polymers [12,13,14,15,16,17,18]. The combination of biodegradable polyesters with substances of plant origin and their derivatives can allow to obtain a new generation of polymeric materials that will be fully degradable and safe for consumers [19,20].

The aim of study was obtained pro-ecological, biodegradable polyester materials containing gallates. Chemically, gallates are alkyl esters of 3,4,5-trihydroxybenzoic acid, the naturally occurring gallic acid. Gallic acid is one of the most abundant phenolic acids in the plant kingdom. It has been isolated from different plants such as oaks and pomegranates. Due to its free-radical scavenging and antioxidant properties, gallic acid and its derivatives, such as lauryl gallate, propyl gallate, octyl gallate, tetradecyl gallate, and hexadecyl gallate, can inhibit oxidation and rancidity in oils and fats [21,22,23,24,25,26]. Gallates are antioxidants used as preservatives in the food, cosmetics, and pharmaceutical industries. In industry, synthetic derivatives of gallic acid, i.e., propyl gallate (E310), octyl gallate (E311), and dodecyl (lauryl) gallate (E312), are commonly used [27,28].

In this manuscript, gallates were used as environmentally and consumer-friendly stabilizers to control the lifetime of biodegradable polyesters. Such an application has not been described in the literature so far and is a scientific novelty. It seems that the proposed materials can be successfully used as packaging materials that will not be deposited in landfills. Gallates are widely used as food preservatives, which is why polyester materials with the addition of these compounds should be safe for human health.

## 2. Materials and Methods 

### 2.1. Reagents

The polymers used for the study were biodegradable aliphatic polyesters: PLA and polymer P(3,4HB) 2001 from the PHA group of polymers. PLA (IngeoTM Biopolymer 4043D PLA) was bought from Nature WorksTM (Minnetonka, MN, USA) and had the following properties: *T*_g_ = 55–60 °C, *T*_m_ = 145–160 °C, and Melt Flow Index MFI = 6 g/10 min; PLA contained 4.8 % D-lactide and had an average molecular weight (*M*_w_) of 200 kDa. PHA was obtained from Simag Holdings LTD (Hong Kong, China) and had the following properties: P(3,4HB) containing 12 mol % 4-hydroxybutyrate, average *M*_w_ of approximately 520 kDa, MVR = 15–20 g/10 min, (assay conditions: temperature 170 °C, nominal load 2.16 kg), and density of 1.25 g/cm^3^. The configuration of the 3HB component in the copolymer was R-3HB.

Gallates (ethyl gallate *M*_w_ = 198.17 g/mol, antioxidant, ≥96.0% (HPLC); propyl gallate *M*_w_ = 212.20 g/mol, antioxidant, ≥98.0% (HPLC); octyl gallate *M*_w_ = 282.33 g/mol, antioxidant, ≥99.0% (HPLC), and lauryl gallate *M*_w_ = 338.44 g/mol, antioxidant, ≥99.0% (HPLC)) were obtained from Sigma Aldrich (Steinheim, Germany). Figure 1 shows the structural formulas of the polymers and gallates used.

### 2.2. Preparation of PLA and PHA Samples Containing Gallates

Granules of PLA and PHA were dried (12 h, 55 °C) and mixed with 1 part by weight of gallates. The mixtures were extruded using a laboratory extruder. Strip samples with a thickness of 1.6–1.8 mm were obtained. The temperature of the working chamber of the extruder was 180 °C for PLA and 160 °C for PHA, the screw rotation speed was 40 rpm, and the extrusion pressure was 17 atm.

### 2.3. Analysis of the Properties of Gallates

#### 2.3.1. Antioxidant Activity Measured by the ABTS and DPPH Methods

The antioxidant properties of gallates were determined by reduction of the free radicals 2,2′-azino-bis(3-ethylbenzothiazoline-6-sulphonic acid) ABTS and 2,2-diphenyl-1-picrylhydrazyl DPPH. The inhibition level (%) of ABTS and DPPH was calculated according to the following Equation (1):Inhibition (A%) = [((A_0_ − A_1_) /A_0_) 100](1)
where *A*_0_ is the absorbance of the control sample without antioxidants, and *A*_1_ is the absorbance in the presence of gallates. The authors described fully ABTS and DPPH methods in other publications [29,30].

#### 2.3.2. Determination of Ion Reduction by the FRAP and CUPRAC Methods

The ability of gallates solutions to reduce the ferric ion (Fe^3+^→Fe^2+^) under acidic conditions was determined using the ferric reducing antioxidant power (FRAP) assay. The cupric reducing antioxidant capacity (CUPRAC) method is analogous to the FRAP determination and consists in the reduction of Cu^2+^ to Cu^1+^. A detailed description of the methods is presented in other publications [29,30]. 

The ferric (FRAP) and cupric (CUPRAC) ions reducing power was calculated as follow (2):Δ*A* = *A*_AR_ − *A*_0_(2)
where *A*_0_ is the absorbance of the tested reagent, and *A*_AR_ is the absorbance after the reaction.

Absorbance measurements using the ABTS, DPPH, FRAP, and CUPRAC methods were performed using a UV spectrophotometer (Evolution 220, Thermo Fisher Scientific, Waltham, MA, USA).

#### 2.3.3. Thermal Stability of Gallates

Thermal decomposition (TGA) was determined using a Mettler Toledo Thermobalance (Greifensee, Switzerland) to obtain the initial temperature of degradation and the maximum thermal degradation temperature. Gallates samples with a weight of 5 mg were heated at a 10 °C/min from 25 to 800 °C under an inert N_2_ atmosphere.

### 2.4. Analysis of PLA and PHA Samples Containing Gallates

#### 2.4.1. Specific Migration of Gallates from Polymers

Tests of specific migration of gallates were carried out according to the European Standard EN 13130. The release of gallates from the polymers was studied in a model solution of ethanol at the temperature of 25 °C. Samples of extruded strips with a thickness of 1.6–1.8 mm and an area of 1 cm^2^ were immersed completely in 10 ml of ethanol and incubated at 25 °C for 14 days. Volumes of 2 mL of ethanol model solutions in contact with the polyester containing gallates were taken after 2, 4, 7, 9, 11, and 14 days of incubation in order to follow the release of the additive in time. The solutions were scanned from 190 to 1100 nm in a UV spectrophotometer (Evolution 220, Thermo Fisher Scientific, Waltham, MA, USA). Ethanol was used as a blank.

#### 2.4.2. Scanning Electron Microscopy (SEM)

Based on the images obtained from the scanning electron microscope (SEM) LEO 1530 (Carl Zeiss AG, Germany), the morphology of the samples was evaluated. The test samples were prepared as follows: samples were fractured in liquid nitrogen and sputtered with carbon. Magnification was 1000×.

#### 2.4.3. Differential Scanning Calorimetry (DSC)

The temperature ranges of the samples’ phase changes were determined using the Mettler Toledo DSC analyser (TA 2920; TA Instruments, Greifensee, Switzerland). The samples of polymers (5–6 mg, placed in 100 μL-volume open aluminium crucibles) were heated from 0 to 200 °C at a rate of 20 °C/min under an argon atmosphere. After 10 min at 200 °C, the samples were cooled to 0 °C. Then, the gas was switched from argon to air (flow rate 50 mL/min), and the samples were heated to 350 °C. The following parameters were tested: glass transition temperature (*T*_g_), crystallization temperature (*T*_cc_), melting temperature of the crystalline phase (*T*_m_), and oxidation temperature (*T*_o_). The heat (Δ*H*) accompanying the phase changes was also determined. 

#### 2.4.4. Wide-Angle X-ray Diffraction (WAXD)

An X-ray diffractometer, Bruker model D2 Phaser (Billerica, MA, United States), equipped with a Lynxeye detector, was used to analyze the crystal structure. The tests were performed in the 2θ angle range of 2–40° with steps of 0.02°.

#### 2.4.5. Mechanical properties

Polymers’ mechanical properties were analyzed using a Zwick Roell Z005 test machine (manufacturer Zwick Roell, Germany) before and after ageing. To determine the mechanical properties, six test samples were cut out from extruded trips with a thickness of 1.6–1.8 mm and length of 150 mm. The measurement conditions were: a preload of 0.1 N and a test speed of 50 mm/min. The parameters tensile strength (MPa) and elongation at break (%) (*E*_b_) were determined. 

#### 2.4.6. Surface Free Energy of Polyester Samples

The OEC 15EC goniometer (DataPhysics Instruments GmbH, Filderstadt, Germany) was used to determine the surface free energy of polymers before and after ageing. The surface free energy was calculated by the method of Owens, Wendt, Rabel, and Kaelble (OWRK), using software module SCA 20. Polar and disperse contributions to the surface energy and surface tension, respectively, were combined. Therefore (3,4):(3)$$\sigma_{l}=\sigma_{l}^{d}+\sigma_{l}^{p}$$
(4)$$\sigma_{S}=\sigma_{S}^{d}+\sigma_{S}^{p}$$
where: $\sigma_{l}^{d}$ and $\sigma_{l}^{p}$ represent the disperse and polar parts of the liquid, while $\sigma_{S}^{d}$ and $\sigma_{S}^{p}$ stand for the respective contributions of the solid.

Surface energy measurements were made based on the determination of contact angles. The contact angles were measured for liquids with different polarities: distilled water, diiodomethane, and ethylene glycol. During the determination of surface energy for each of the three samples of one material, 10 contact angles were made for each of the three liquids. 

#### 2.4.7. Change of Color After Ageing

The change of color was measured using a CM-3600d spectrophotometer (Konica Minolta Sensing, Japan) after ageing of the samples (100, 200, and 300 h of UV exposure, thermaloxidation, and weathering). The test provides the color as described in the CIE-Lab space and in a system of three coordinates: *L*, *a*, and *b*, where *L* is the lightness parameter (maximum value of 100, representing a perfectly reflecting diffuser, minimum value of zero, representing the color black), a is the axis of red–green, and *b* is the axis of yellow–blue. The *a* and *b* axes have no specific numerical limits. The change of color, dE*ab, was calculated according to following Equation (5):(5)$$\mathrm{dE}*\mathrm{ab}=\sqrt{\left(\Delta a^{2})+(\Delta b^{2})+(\Delta L^{2}\right)}$$

For change of color measurements, 3 samples of each type of material were prepared for each time interval of aging. 

### 2.5. Controlled Aging of Polyesters

All aging tests was performed at intervals of 100, 200, and 300 h.

#### 2.5.1. UV Exposure

UV ageing was performed using a UV 2000 apparatus (Atlas Material Testing Technology LLC, Mt Prospect, IL, USA). The measurement lasted 100, 200, and 300 h and consisted of two alternately repeating cycles characterized by daily cycle (radiation intensity = 0.7 W/m^2^, temperature 60 °C, and duration 8 h) and night cycle (no UV radiation, temperature = 50 °C, and duration 4 h).

#### 2.5.2. Thermooxidation Aging

The samples were exposed to air at an elevated temperature (70 °C) for 100, 200, and 300 h in a dryer (Binder, Germany) with forced convection.

#### 2.5.3. Weathering Aging 

This type of aging was analyzed using a Weather-Ometer Ci 4000 (Atlas Material Testing Technology LLC, USA) with inner and outer filters of type S borosilicate glass. The test consisted of two variable cycles simulating daytime and nighttime conditions, and the samples were subjected to two different cycles as follows: day cycle (radiation intensity E = 40 W/m^2^ = 0.144 MJ/m^2^ over a λ range of 300–400 nm, temperature of 60 °C, duration of 240 min, humidity at 80%, rain water on) and night cycle (no radiation, temperature at 50 °C, humidity of 60%, duration of 120 min). Material weathering tests with a xenon-arc light were performed according to ISO 4892-2 (accelerated weathering simulates the damaging effects on materials and coatings of long-term outdoor exposure).

## 3. Results and Discussion

The first part of the study was to examine the properties of gallates, including their ability to stabilize polymers and their thermal stability. The ability to stabilize polymeric materials was determined based on the antioxidant activity of gallates and their ability to reduce transition metal ions. Investigating the thermal stability of gallates allowed the assessment of gallate resistance to degradation at polymers’ processing temperatures. 

Figure 2 summarizes the antioxidant properties of gallates measured by the ABTS and DPPH methods, as well as their ability to reduce iron and copper ions determined by the FRAP and CUPRAC methods. Solutions at concentrations of 0.01–0.1 mol/mL of all gallates showed significant properties that may improve the stability of polymer materials. The ability to reduce free radicals (ABTS and DPPH methods) and the ability to reduce transition metal ions (FRAP and CUPRAC methods) increased with the increase of the concentration of all gallates solutions, reaching very high values, e.g., ethyl gallate (C = 0.06 mol/mL) showed ability to reduce ABTS radicals equal to 97.6% and ability to reduce DPPH radicals equal to 94.1%. 

The antioxidant activity and the ability to reduce transition metal ions of all analyzed gallates changed in the same manner: as the concentration of gallate increased, their antioxidant capacity increased. Sometimes the difference between the oxidation activities of individual concentrations of the test compound was small, e.g., DPPH activities of lauryl gallate at 0.03 and 0.06 mol/mL were similar. These were not extraordinary changes. This behavior, during ABTS, DPPH, FRAP and CUPRAC determinations, is quite typical for substances of natural origin.

The effect of molar mass and gallate substitution on their antioxidant properties was also interesting. The gallates with lower molar mass (ethyl and propyl gallate) had higher antioxidant properties and greater ability to reduce Fe^3+^ and Cu^2+^ ions than the gallates with higher molar mass (octyl and lauryl gallates). The presence of large substituents in the structure of octyl and lauryl gallate is a spatial hindrance that reduces the activity of these compounds. 

The high antioxidant activity of all gallates should stop unfavorable oxidation processes associated with the degradation of polymeric materials. Transition metal ions can catalyze the aging reactions of polyesters, so the good ability of gallates to reduce transition iron and copper ions is an important property in the stabilization of polymers.

Before polymers processing, the thermal stability of gallates additives was analyzed. Figure 3 and Table 1 show the results of thermogravimetry of gallates. Table 1 shows the values of T20, T50, and T90 for the gallates, which refer to the loss of 20%, 50%, and 90%, respectively, of the initial mass of the sample as a function of temperature. Lower molecular weight gallates, i.e., ethyl and propyl gallates, were decomposed in three steps, while octyl gallate was decomposed in two steps, and the highest molecular weight gallate (lauryl gallate) was decomposed in one step. All gallates were characterized by high thermal stability. Extrusion of biodegradable polyesters was carried out at temperatures of 160 and 180 °C, i.e., below the decomposition temperature of gallates. The T20 of gallates ranged from 273 to 318 °C. 

In the next part of the study, samples of polyesters containing gallates were characterized. The SEM images in Figure 4 show the morphology of the polymeric materials. The PLA samples were more compact and smoother than the PHA samples. As opposed to samples based on PLA, PHA had a more porous structure. Gallate is clearly visible in the PLA matrix in the SEM image (marked with a red circle), but it is hardly visible in the SEM photograph of PHA/octyl gallate. 

The morphology of polymeric materials is related to the ability of gallates to migrate from the polyester samples (Figure 5). Ethanol was chosen as a model solution because gallates are well soluble in this solvent. Due to this choice of solvent, migration from the polyesters was clearly visible. Gallates migrated to a greater extent from PHA because of its a porous structure. The compact structure of PLA significantly reduced the migration of gallates from the samples. The concentrations of migrant gallates from PHA were at least two times greater than those from PLA. Lower molar mass gallates (ethyl and propyl gallate) migrated more than octyl and lauryl gallates. The higher molar mass and the presence of spatial hindrance reduced the mobility of these latter gallates and their migration from polyester materials. The largest migration was observed within seven days of measurements, after which time gallates’ concentration in the model liquid stabilized.

Gallates are added to food as stabilizers. A stabilizing effect was also found for polymeric materials. Table 2 presents the results of differential scanning calorimetry tests. The DSC curves of the PHA materials present two peaks corresponding to the melting of the crystalline phase; therefore, there are two values, *T*_m_ and Δ*H*_m_, in Table 2 for a single sample. The addition of gallates did not change the temperature ranges of the samples’ phase changes, i.e., glass transition temperature (*T*_g_), crystallization temperature (*T*_cc_), and melting temperature (*T*_m_). However, the addition of gallates caused an increase in the oxidation temperature of both polyesters. The oxidation temperature is the temperature at which an exothermic oxidation peak appears in the DSC curve. Table 2 presents the initial (onset) and final (endset) temperatures of the oxidation peaks. A significant increase in the oxidation temperature (*T*_o_) was found for PLA containing ethyl, propyl, and octyl gallate, corresponding, respectively to about 50, 61, and 45 °C. For PHA containing ethyl, propyl, and octyl gallate, an increase in the initial oxidation temperature of about 17, 26, and 26 °C, respectively, was found. Furthermore, for both polyesters, the lowest increase in oxidation temperature was observed for lauryl gallate samples: of about 13 °C for PLA, and of about 11 °C for PHA. In addition, the presence of gallates in the polymers increased the final oxidation temperature of all samples. Higher oxidation temperatures of the PLA and PHA samples containing gallates should increase the stability of the materials and improve their resistance to oxidation. The results suggest that the analyzed gallates can be successfully used as effective stabilizers for biodegradable polyester materials.

Based on the DSC analysis, the degree of crystallinity of the polyesters containing gallates was characterized. The crystallinity of polymeric materials can be calculated by DSC, knowing the melting enthalpy of the 100% crystalline 100% polymer. The degree of crystallinity, *X*_c_, was calculated as follows: *X*_c_(%) = (∆*H*_m_ − ∆*H*_cc_)/∆*H*_0_ × 100, where ∆*H*_m_ is the melting enthalpy of the test polymer, Δ*H*_cc_ is the enthalpy of crystallization of the samples, and ∆*H*_0_ is the melting enthalpy of the 100% crystalline 100% PLA or PHA. In accordance with the literature, the melting enthalpy estimated for 100% crystalline PLA is 97.3 J/g [31]. According to the producer, the PHA polymer is P(3,4HB), so the melting enthalpy of 100% crystalline PHB of 146 J/g [32] was used as the melting enthalpy of 100% crystalline PHA. The value of melting enthalpy of 100% crystalline PHB does not quite correspond to the value of melting enthalpy of 100% crystalline P(3,4HB), and the calculations contain a certain error resulting from these differences. Nevertheless, the only value available in the literature of the melting enthalpy of 100% crystalline PHB is 146 J/g, indicated as melting enthalpy for polymers from the PHAs group. The DSC curves of the PHA samples had two peaks corresponding to the melting of the crystalline phase; therefore, the melting enthalpy of the PHA samples used for the calculation of crystallinity was the sum of the Δ*H*_m_ of the two peaks corresponding to the melting of the crystalline phase. PLA is an amorphous polymer. The calculated degree of crystallinity was 0% for *X*_cPLA_ and 0% for PLA containing gallates. The addition of gallates to PLA did not affect the crystallinity of the material. In contrast, the PHA samples were crystalline. The degree of crystallinity of the reference sample was 17.1% and, after the addition of gallates, remained at the same level or decreased slightly. The addition of gallates to crystalline PHA could disturb the polymer structure and reduce the degree of crystallinity of the material. The calculated content of the crystalline phase was consistent with the WAXD results.

Figure 6 shows the WAXD spectra of reference PLA and PHA and of the samples containing octyl gallate. The WAXD spectrum of the PLA samples is related to the amorphous nature (DSC results – *X*_cPLA_ = 0% and 0% for PLA/octyl gallate) of the materials. On the WAXD diffractogram for the PLA samples, only broad maxima of diffusive scattering occurred, which indicated the existence of only close-range ordering in this polymer at the molecular level and thus the amorphous internal structure of the samples. The WAXD spectrum also confirmed the crystalline nature of the PHA-based samples (*X*_cPHA_ = 17.1%, and PHA/octyl gallate 16.3%, calculated on the basis of the DSC results). The sharp diffraction peaks (the most pronounced, with the highest intensity at 16.3°, and two smaller peaks at 13.0° and 18.2°) found in the WAXD spectra showed that the reference PHA was a crystalline polymer. Adding octyl gallate to PHA caused a decrease in the crystallinity of the material, as evidenced by the smaller number of sharp diffraction peaks found in the WAXD spectra. In the WAXD spectra of PHA/octyl gallate, only one peak was observed at 12.9°. The PHA/octyl gallate sample had a higher degree of disorder (greater amorphism) than the pure PHA sample. The addition of octyl gallate to the PHA polymer caused a disruption of the polymer matrix structure and the appearance of short-range ordering in this polymer at the molecular level.

In the next step, the effect of controlled aging on the mechanical properties of the samples, surface energy, and color change was analyzed. Figure 7 and Figure 8 summarize the changes in PLA and PHA mechanical properties (*T*_s_, the tensile strength and E_b_, elongation at break) before and after thermooxidation and UV aging. 

The addition of gallates to PLA caused a slight decrease in the tensile strength (*T*_s_ of PLA with gallates = 47.6–50.7 MPa) relative to that of the reference sample (*T*_s_ = 51 MPa) and also an increase in the elongation at break (*E*_b_ of PLA = 4.2%; *E*_b_ of PLA with gallates = 4.5–5%). Unlike PLA, the addition of gallates to PHA caused a slight increase in *T*_s_ (*T*_s_ of PHA = 5.6 MPa; *T*_s_ of PHA with gallates = 5.9–9.0 MPa) and a marked decrease in *E*_b_ (*E*_b_ of PHA = 24.4%; *E*_b_ of PHA with gallates = 9–14.6%). The addition of gallates to polyester materials caused contrasting changes in the mechanical properties of the samples, due to the different nature of the materials (PLA is amorphous, and PHA is crystalline).

Thermooxidation and UV aging of polylactide-based samples resulted in an increase in the tensile strength and a decrease in the elongation at break after 100 h of aging. Similar results were obtained for materials made of PHA. After the initial thermooxidation and UV aging, polymer crystallization occurred, resulting in an increase in the tensile strength and a decrease in the elongation at break. After 200 h of both types of aging, further material degradation, due to the deterioration of the mechanical properties (i.e., decrease of *T*_s_ and *E*_b_), took place. The changes in the mechanical properties were partly due to the crystallization of the two polyesters.

Figure 9 shows the surface free energy of the polyester samples before and after weathering aging. The addition of gallates to both polymers did not significantly affect their energies, which were comparable to those of the reference samples. Weathering aging caused an increase in the surface free energy of the polyesters due to degradation of the polymer matrix. The surface free energy of PLA after 300 h of weathering aging increased compared to that of the reference samples from about 8% (PLA/lauryl gallate) to 31% (PLA/ethyl gallate), and for the PHA materials, it increased by about 7% (PHA/propyl gallate) to 23% (PHA/lauryl gallate).

The color change of polymers is the first sign of their degradation. Figure 10 and Figure 11 show the color changes of the PLA and PHA samples after weathering, thermoxidation, and UV aging. Statistically, when the change of the color coefficient dE * ab is 2 < dE * ab < 3.5, a difference of color can be seen by the average observer. Values of dE * ab in the range of 3.5 < dE * ab < 5.0 mean that there is a distinct color difference, while if dE * ab > 5, colors are perceived as completely different. Thus, the PLA and PHA samples (excluding the reference samples after UV exposure) were characterized by a significant and very pronounced color change. The smallest color change coefficients for both types of samples were found after thermooxidation aging. After weathering and UV aging, very slight changes in the color of the PLA and PHA reference samples and very clear changes in the color of the polymers containing gallates were observed.

Figure 10D and Figure 11D collate photos of samples before aging and during exposure to UV light. The addition of gallates did not change the color of the samples of both polyesters. Gallates dissolved well in polymers, and both types of samples were homogeneous (PLA was transparent, and PHA was milky-white). UV aging, as well as other types of aging, caused the color of PLA to change from transparent to insignificantly milky, and that of PHA to change from milky-white to slightly yellow. Under the influence of all types of aging, the samples of both polyesters containing gallates turned yellow, which was particularly pronounced for the transparent materials based on PLA. A clear change in color during aging can be an indicator of the lifetime of materials.

## 4. Conclusions

Ethyl, propyl, lauryl, and octyl gallates demonstrated strong antioxidant properties and very good ability to reduce transition metal ions. In addition, all gallates were characterized by high thermal stability, higher than that of biodegradable polyesters. Due to these properties, gallates can be successfully used as polymer stabilizers, as evidenced by their high oxidation temperature. The addition of an appropriate amount of gallates to polyesters can allow the regulation of the lifetime of a polymer material and its controlled degradation. Moreover, the analyzed gallates changed color under the influence of various external factors (UV radiation, temperature, humidity), which resulted in a change in the color of the samples. A change in the color of polymeric materials may be indicative of their lifetime. Gallates, especially those with a low molecular weight (ethyl and propyl), showed the ability to migrate from polymers (mainly those with a porous structure). Migrating gallates, which are commonly used as food preservatives, can protect packaging products and contribute to adding smaller amounts of other preservatives directly to them. Biodegradable polyesters with the addition of gallates can be an environmentally friendly alternative to petrochemical packaging materials. 

## Figures and Tables

**Figure 1 polymers-12-00677-f001:**
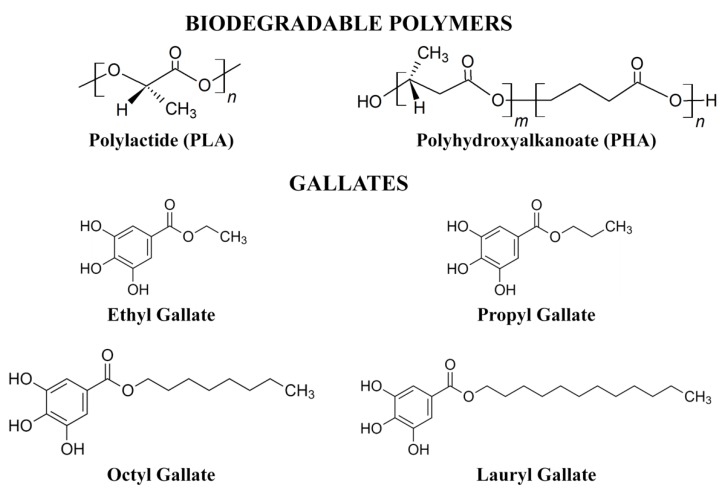
Structure of biodegradable polymers and gallates.

**Figure 2 polymers-12-00677-f002:**
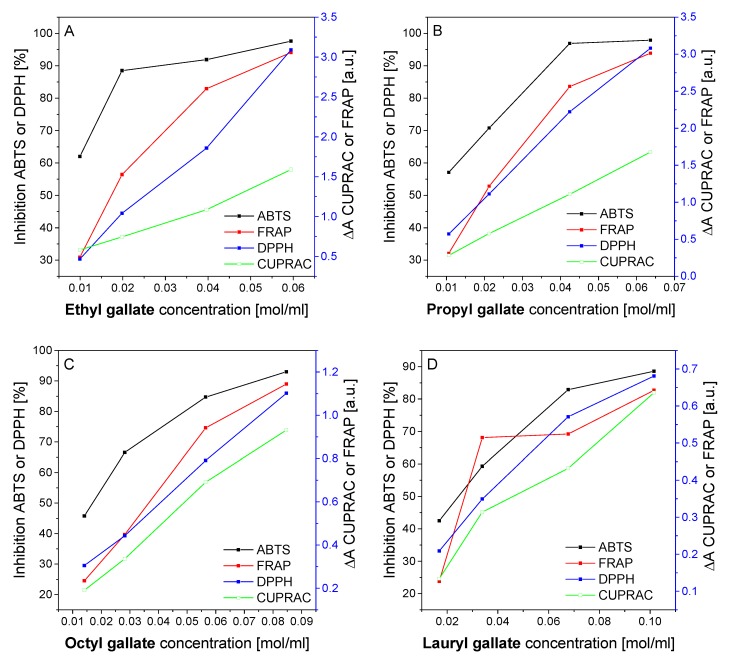
Antioxidant properties of ethyl gallate (**A**), propyl gallate (**B**), octyl gallate (**C**), and lauryl gallate (**D**) measured by the ABTS and DPPH methods and their ability to reduce iron (ferric reducing antioxidant power, FRAP) and cupric ions (CUPRAC).

**Figure 3 polymers-12-00677-f003:**
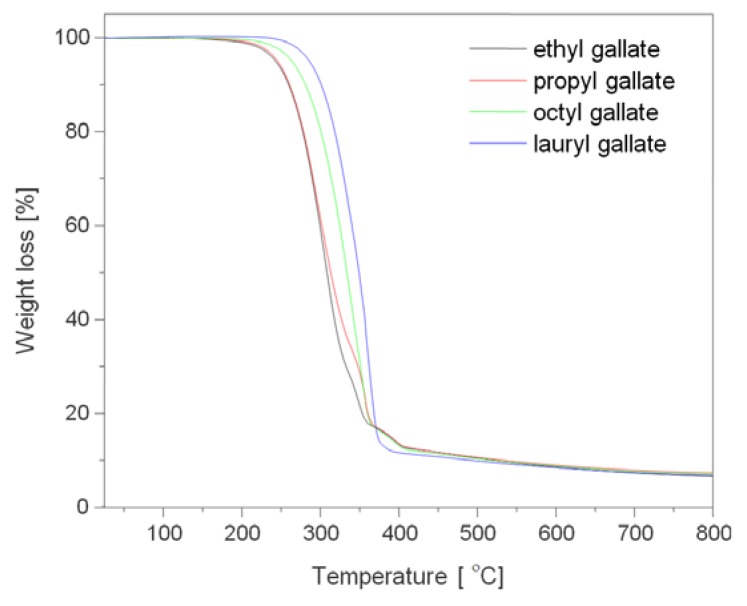
Thermal decomposition (TGA) curves of gallates.

**Figure 4 polymers-12-00677-f004:**
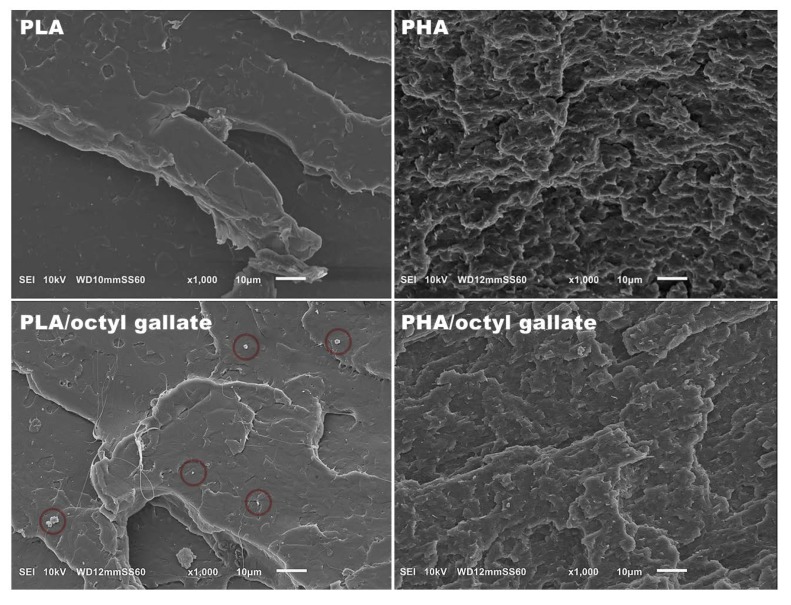
Scanning electron microscope (SEM) photographs of polyesters.

**Figure 5 polymers-12-00677-f005:**
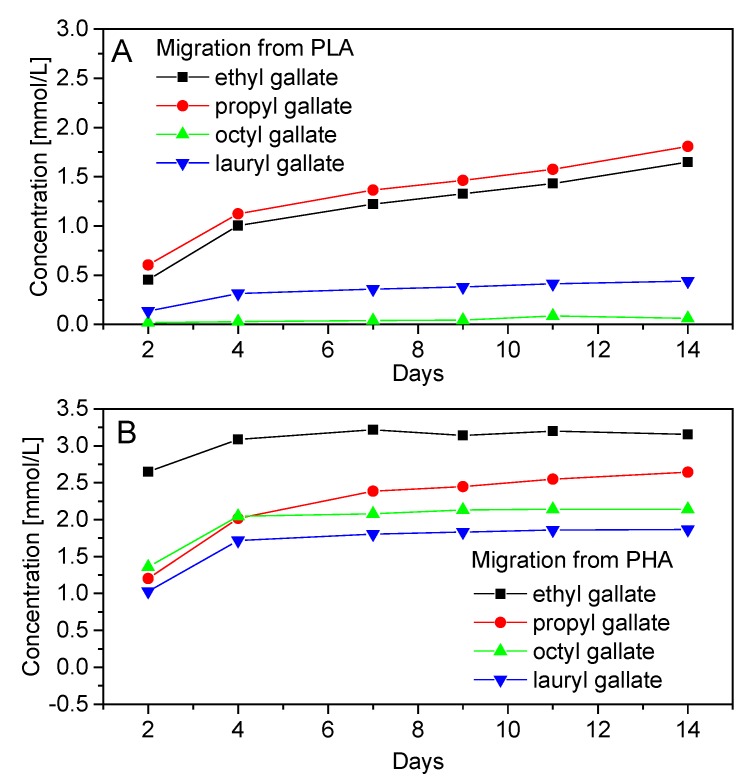
Migration of gallates from PLA (**A**) and PHA (**B**) during 14 days.

**Figure 6 polymers-12-00677-f006:**
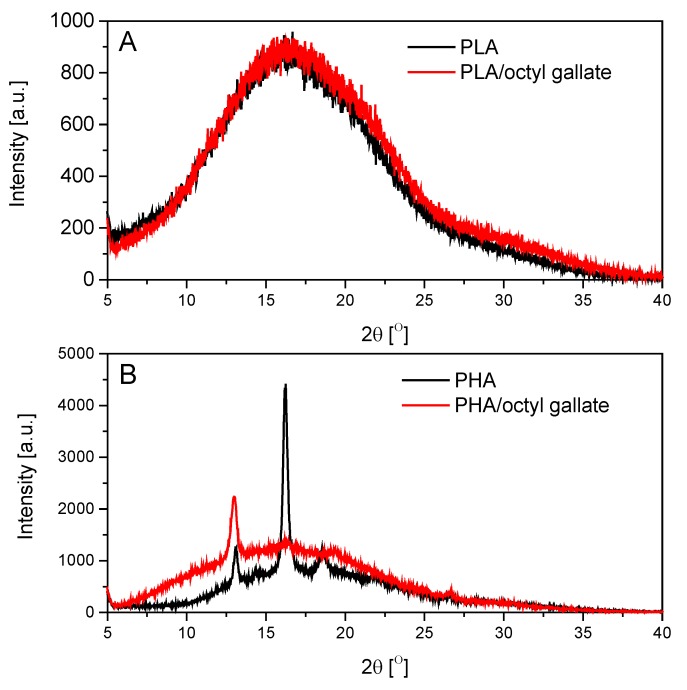
Wide-angle X-ray diffraction (WAXD) spectra of PLA (**A**) and PHA (**B**) samples with gallates.

**Figure 7 polymers-12-00677-f007:**
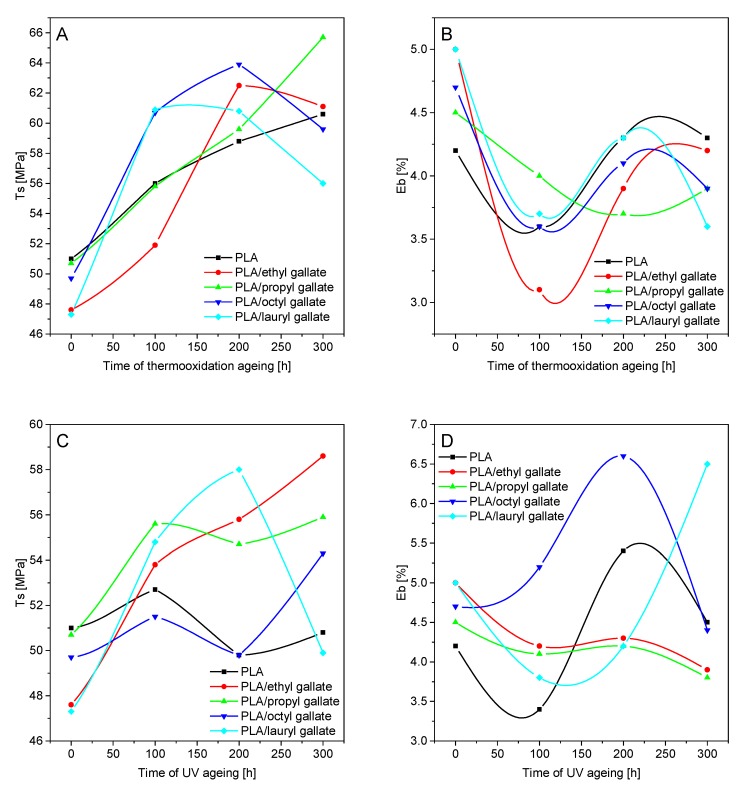
Changes in PLA mechanical properties (*T*_s_: tensile strength and *E*_b_: elongation at break) before and after thermooxidation (**A**,**B**) and UV aging (**C**,**D**).

**Figure 8 polymers-12-00677-f008:**
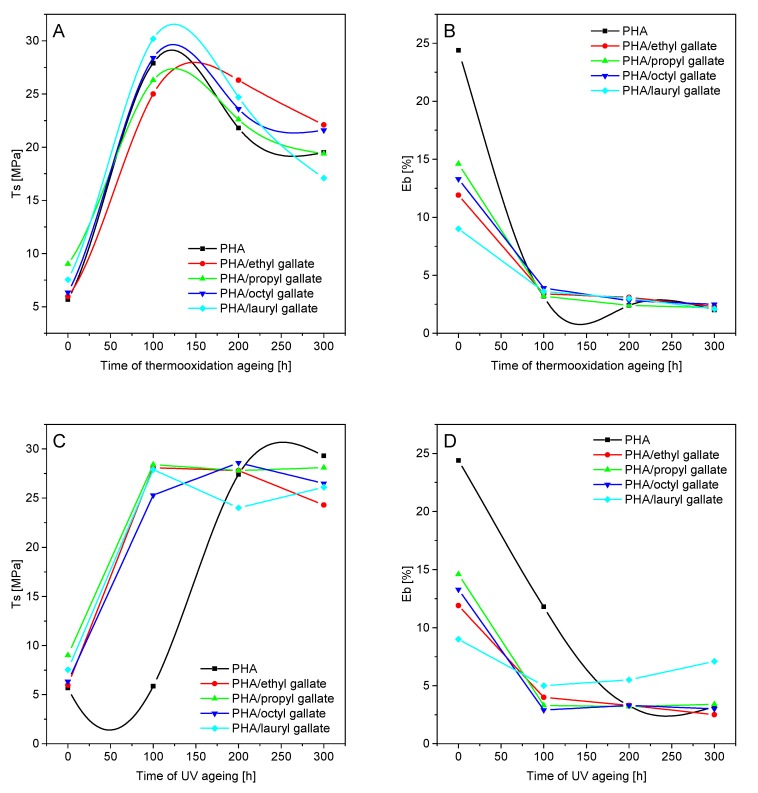
Changes in PHA mechanical properties before and after thermooxidation (**A**,**B**) and UV aging (**C**,**D**).

**Figure 9 polymers-12-00677-f009:**
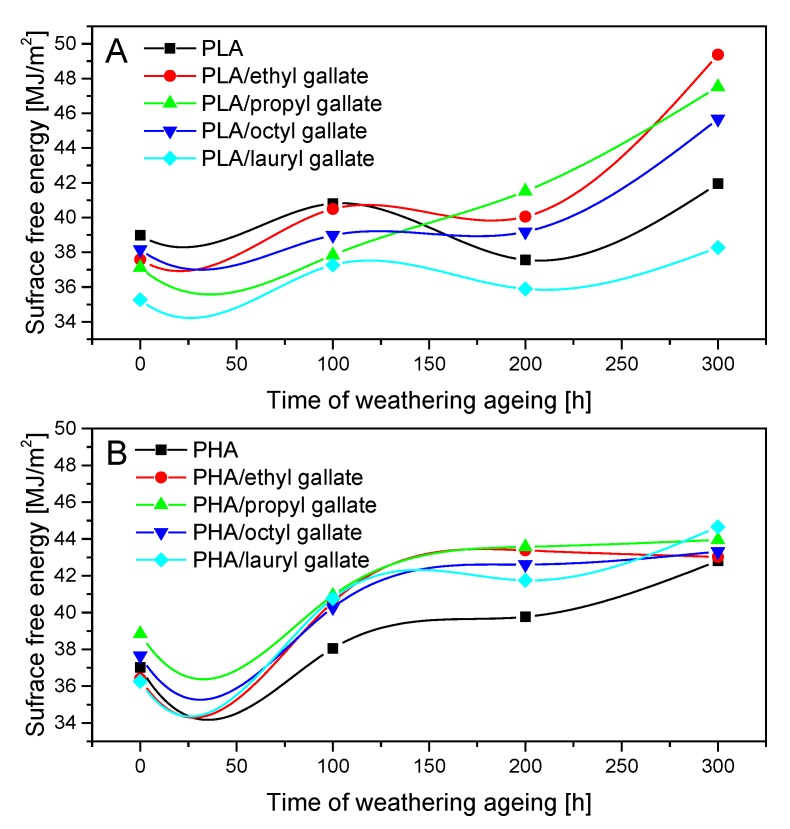
Changes in PLA (**A**) and PHA (**B**) surface free energy after weathering aging.

**Figure 10 polymers-12-00677-f010:**
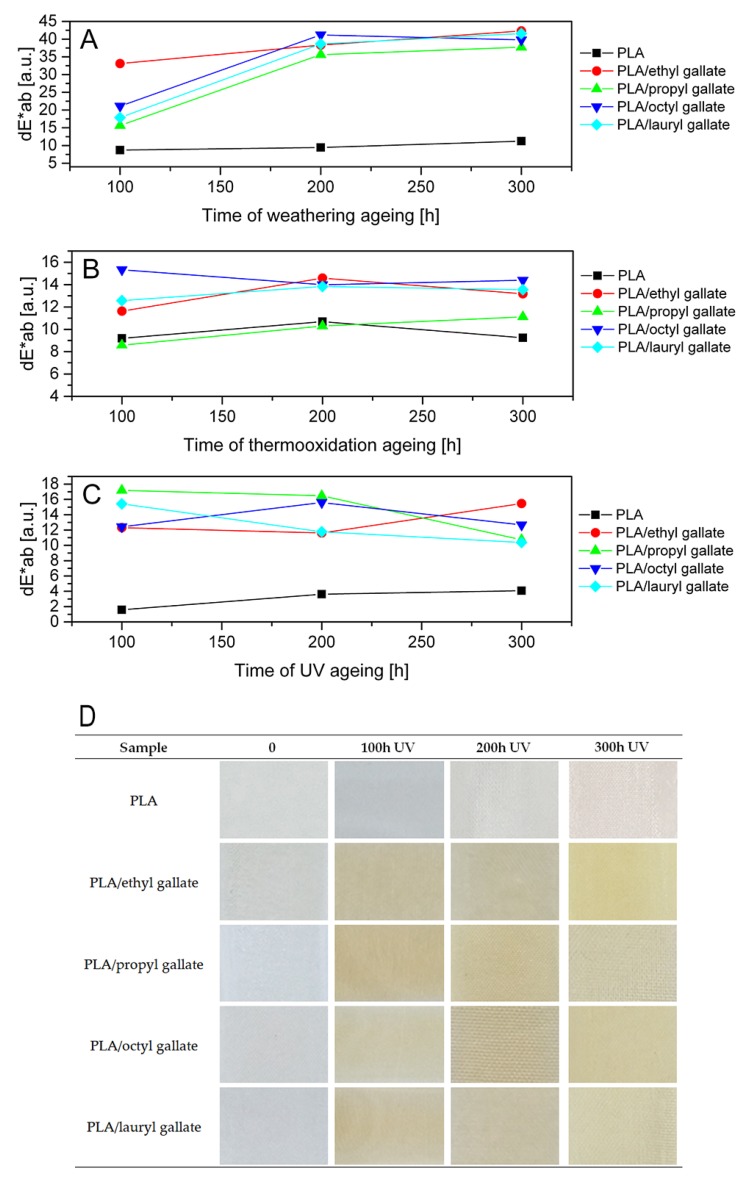
Changes in color of PLA after weathering (**A**), thermooxidation (**B**), and UV (**C**) aging. Visual change of color of PLA containing gallates before and after UV aging (**D**).

**Figure 11 polymers-12-00677-f011:**
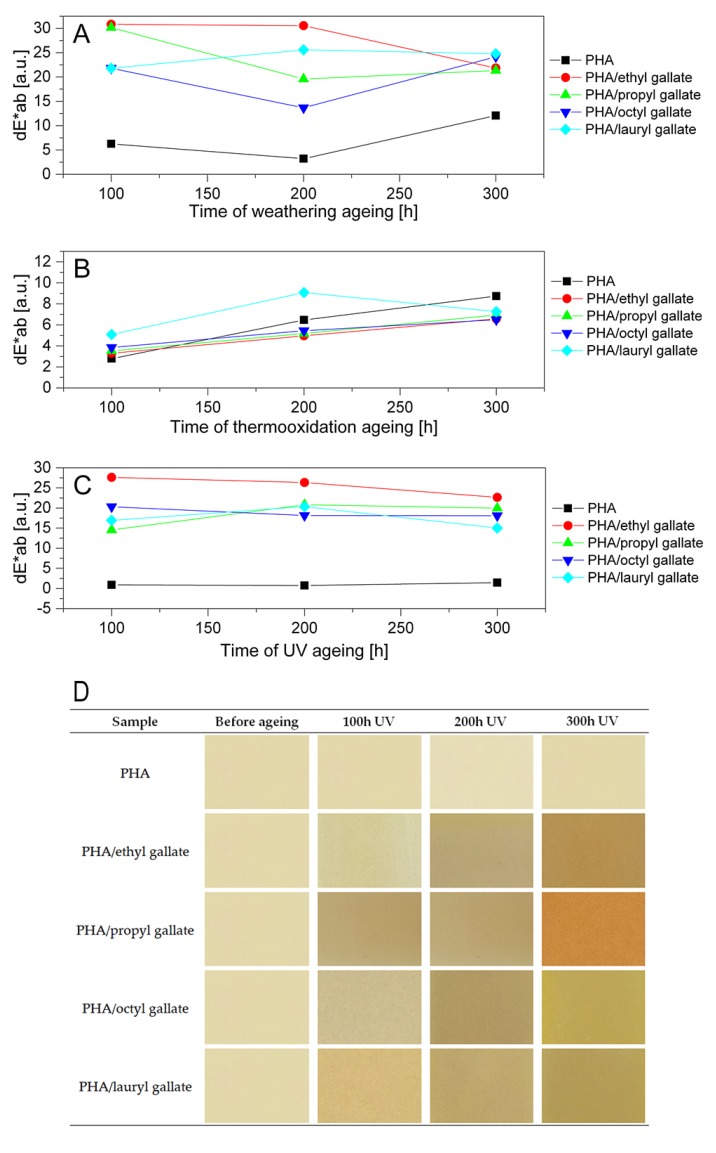
Changes in color of PHA after weathering (**A**), thermooxidation (**B**), and UV (**C**) aging. Visual change of color of PHA containing gallates before and after UV ageing (**D**).

**Table 1 polymers-12-00677-t001:** T20, T50, and T90 of the examined gallates.

Sample	T20	T50	T90
ethyl gallate	277	308	478
propyl gallate	273	309	483
octyl gallate	300	333	468
lauryl gallate	318	349	433

**Table 2 polymers-12-00677-t002:** DSC analysis of the PLA and PHA samples containing gallates.

Sample	*T*_g_/°C	Δ*H*_cc_/J/g	*T*_cc_/°C	Δ*H*_m_/J/g	*T*_m_/°C	Δ*H*_o_/J/g	*T*_o_/°C	*X*_c_/%
PLA	58.4	4.6	108.6	3.1	145.6	22.6	226.2 295.7	0
PLA/ethyl gallate	57.0	5.1	105.8	2.1	145.0	5.3	276.0 305.7	0
PLA/propyl gallate	57.3	4.4	110.0	2.8	144.7	11.4	287.2 302.7	0
PLA/octyl gallate	59.9	7.8	105.2	5.4	144.1	57.9	270.8 315.4	0
PLA/lauryl gallate	59.6	16.3	101.0	14.0	141.9	17.5	238.8 316.5	0
PHA	36.7	15.2	76.5	(1) 6.5 (2) 33.9	(1) 127.9 (2) 156. 4	9.7	199.1 259.3	17.1
PHA/ethyl gallate	39.9	14.5	80.2	(1) 5.2 (2) 31.6	(1) 127.9 (2) 156.1	8.5	216.9 263.5	15.2
PHA/propyl gallate	38.4	13.5	80.0	(1) 6.3 (2) 33.1	(1) 127.7 (2) 156.7	11.4	225.0 267.4	17.7
PHA/octyl gallate	39.9	13.5	80.7	(1) 5.1 (2) 32.3	(1) 127.5 (2) 156.5	25.4	225.0 274.1	16.3
PHA/lauryl gallate	39.9	16.6	81.5	(1) 6.2 (2) 33.8	(1) 123.1 (2) 156.1	22.0	210.2 269.9	16.1

*T*_g_: glass transition temperature, Δ*H*_cc_: enthalpy of crystallization, *T*_cc_: crystallization temperature, Δ*H*_m_: enthalpy of melting, *T*_m_: melting temperature, Δ*H*_o_: enthalpy of oxidation, *T*_o_: oxidation temperature (initial and final), *X*_c_: degree of crystallinity.

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
