# Peer review of "Biodegradable Polyester Materials Containing Gallates"

_polymers, 2020, doi:10.3390/polym12030677_

Round 1
Reviewer 1 Report
This paper reports the effects of gallates on various properties. There are some experimental issues (design and procedures), which decrease the value of this paper. For example, the statements on page 11 "the first stage of degradation of PLA materials was crystallization of the polymer matrix", which indicate that the crystallization of samples before degradation test should be performed. The crystallization during various degradation tests in Figures 7-11 affects the results of properties. That is, in most tests, the authors observed the effects of crystallization, not those of degradation.
(1) PLA and P(3,4HB) are crystallizable at 70°C during thermooxidation ageing, and P(3,4HB) is crystallizable at 50°C during UV exposure and weathering ageing. Crystallization for example at 70-100°C for several hours should be performed before ageing, UV exposure and weathering ageing. Otherwise, "the first stage of degradation of PLA materials was crystallization of the polymer matrix" and results obtained in Figures 7-11 are due to crystallization not degradation.
(2) Antioxidant properties should be compared for concentration in [mol/mL] not in [mg/mL], since the gallate part of molecule have the antioxidant property. So, the gallates with high molecular weight such as lauryl gallate should be the lowest antioxidant property as expected. The Figure 2 should be replotted as a function of concentration in [mol/mL].
(3) The evaluation of crystallinity values in Table 2, the authors neglected the delta Hcc values. However, delta Hm – delta Hcc should be used for evaluation of crystallinity values. If the authors do that, the crystallinity values of all PLLA samples should be zero and are in agreement with WAXD results in Figure 6A.
(4) Unusual changes of propyl gallate in Figure 2A, C and D and of lauryl gallate in Figure 2B has to be explained.
(5) The data for propyl gallate in Figure 3 should be re-measured or explain the reason why this gallate has the highest ash content above 400°C. The seemingly highest ash content of propyl gallate can be the fluctuation of zero point.
(6) Chirality of lactic acid (L- or D-) in PLA and 3HB in P(3,4HB) should be specified. The average molecular weights and polydispersity index values of PLA and P(3,4HB), L-unit content of lactic acid, fraction and D-unit content of 3HB should be specified.
(7) The authors say "the gallate was clearly observed on the SEM image in PLA matrix", which cannot be recognized. The authors are recommended to show the place of gallate with arrows or dotted circles.
(8) The initial values of samples before ageing (at 0 h) is lacking in Figure 10.
Author Response
Institute of Polymer and Dye Technology
Technical University of Lodz
90-924 Lodz, ul Stefanowskiego 12/16, Poland
Tel.: +48 42 631 32 23, Fax: +48 42 636 25 43
March 09, 2020
Professor
Receiving Editor
Polymers (MDPI)
Dear Professor,
We are resubmitting our revised paper entitled “Biodegradable polyester materials containing gallates” by Anna Masek and Malgorzata Latos-Brozio with a request to reconsider it for publication in " Polymers".
We have carefully considered the reviewers' comments. The manuscript was revised exactly according to these comments. The list of responses to the reviewer’s comments and corrections made in the manuscript is attached.
The manuscript has not been previously published, is not currently submitted for review to any other journal, and will not be submitted elsewhere before a decision is made by this journal.
For correspondence please use the following information:
corresponding author: Anna Masek
Institute of Polymer and Dye Technology
Technical University of Lodz
90-924 Lodz, ul Stefanowskiego 12/16, Poland
Tel.: +48 42 631 32 13
Fax: +48 42 636 25 43
e-mail: anna.masek@p.lodz.pl
Reviewer 1
Reviewer 1: This paper reports the effects of gallates on various properties. There are some experimental issues (design and procedures), which decrease the value of this paper. For example, the statements on page 11 "the first stage of degradation of PLA materials was crystallization of the polymer matrix", which indicate that the crystallization of samples before degradation test should be performed. The crystallization during various degradation tests in Figures 7-11 affects the results of properties. That is, in most tests, the authors observed the effects of crystallization, not those of degradation.
(1) PLA and P(3,4HB) are crystallizable at 70°C during thermooxidation ageing, and P(3,4HB) is crystallizable at 50°C during UV exposure and weathering ageing. Crystallization for example at 70-100°C for several hours should be performed before ageing, UV exposure and weathering ageing. Otherwise, "the first stage of degradation of PLA materials was crystallization of the polymer matrix" and results obtained in Figures 7-11 are due to crystallization not degradation.
Answer: Thank you for your important comment. The degradation of biodegradable polymers, especially polylactide, is very specific. According to literature data [1], depending on the degradation conditions, there are different changes in the polylactide crystallinity. These changes also depend on the degree of crystallinity of the polymer. Changes in the polylactide crystallinity are part of the changes associated with the degradation of the polymer material, not a separate process. According to literature, when modelling chain cleavage induced crystallisation in biodegradable PLLAs, it can be assumed that the crystal growth occurs much faster than the hydrolysis reaction. In semi-crystalline PLLAs, the amorphous polymer chains entrapped by the spherulites degrade much faster than the free amorphous polymer chains outside the spherules.
The research results shown in publication [1] suggest that during polylactide degradation at 70°C there is an increase in polymer crystallinity, which accompanies the simultaneous decrease of its molar mass. Polymer crystallization is therefore part of the degradation process.
Based on the literature data [1], as well as our research, we conclude that the results obtained in Figures 7-11 are due to degradation, which is accompanied by changes in the crystallinity of polymers. In this case, crystallization is not a separate process preceding the degradation of polymeric materials. Thank you very much for showing the problem of the specific degradation process of biodegradable polymers, in particular polylactide.
[1] A. Gleadall, J. Pan, H. Atkinson, A simplified theory of crystallisation induced by polymer chain scissions for biodegradable polyesters, Polymer Degradation and Stability, 97 (2012) 1616-1620.
What's more, in another publication [2] it was shown that the addition of natural nucleation agent (increasing the crystallinity of PLA) only slightly accelerates the photodegradation of the material, which is accompanied by cleavage of PLA chains. Moreover, the presence of natural nuleation agents orotic acid in both initial and photodegraded samples was found to influence biodegradation positively by shortening the lag phase and increasing the observed maximal rate of the biodegradation.
[2] J. Salac, J. Šerá, M. Jurca, V. Verney, A. A. Marek, M. Koutný, Photodegradation and Biodegradation of Poly(Lactic) Acid Containing Orotic Acid as a Nucleation Agent, Materials 2019, 12, 481; doi:10.3390/ma12030481.
Reviewer 1: (2) Antioxidant properties should be compared for concentration in [mol/mL] not in [mg/mL], since the gallate part of molecule have the antioxidant property. So, the gallates with high molecular weight such as lauryl gallate should be the lowest antioxidant property as expected. The Figure 2 should be replotted as a function of concentration in [mol/mL].
Answer: Thank you for valuable comment. Figure 2 has been improved. We agree with the Reviewer's considerations – “the gallates with high molecular weight such as lauryl gallate should be the lowest antioxidant property as expected”. Our considerations in the manuscript “The lower molar mass gallates (ethyl and propyl gallate) had higher antioxidant properties and the ability to reduce Fe3+ and Cu2+ ions than higher molar mass gallates (octyl and lauryl gallates).” are in line with the Reviewer's suggestion.
Reviewer 1: (3) The evaluation of crystallinity values in Table 2, the authors neglected the delta Hcc values. However, delta Hm – delta Hcc should be used for evaluation of crystallinity values. If the authors do that, the crystallinity values of all PLLA samples should be zero and are in agreement with WAXD results in Figure 6A.
Answer: The crystallinity of polymeric materials was calculated by DSC and knowledge of the melting enthalpy of 100% crystalline 100% polymer. The degree of crystallinity, Xc, was calculated as follows: Xc(%) = ∆Hm/∆H0*100, where ∆Hm was the melting enthalpy of the test polymer and ∆H0 was the melting enthalpy of 100% crystalline 100% PLA or PHA.
The formula by which we calculated crystallinity is used in the scientific literature for biodegradable polyester samples, including PLA. Example of a publication:
- Vasile, M. Râpă, M. Ștefan, M. Stan, S. Macavei, R. N. Darie-Niță, L. Barbu-Tudoran, D. C. Vodnar, E. E. Popa, R. Ștefan, G. Borodi, M. Brebu, New PLA/ZnO:Cu/Ag bionanocomposites for food packaging, eXPRESS Polymer Letters Vol.11, No.7 (2017) 531–544.
Due to the use of melting entalpy of 100% of crystalline 100% polymers in the calculations, the results may be subject to error. The values used for the calculations are general values, not specific to a specific PLA 4043D and P(3,4)B, which undoubtedly affects the obtained crystallinity values.
Reviewer 1: (4) Unusual changes of propyl gallate in Figure 2A, C and D and of lauryl gallate in Figure 2B has to be explained.
Answer: Antioxidant activity and the ability to reduce transition metal ions of all analyzed gallates change in the same way - as the concentration of gallate increases, the antioxidant capacity increases. Sometimes the difference between the oxidation activity of individual concentrations of the test compound is small, e.g. DPPH activity of lauryl gallate at 0.03 and 0.06mol/ml. However, these are not extraordinary changes. This behavior, during ABTS, DPPH, FRAP and CUPRAC determinations, is quite typical for substances of natural origin.
Reviewer 1: (5) The data for propyl gallate in Figure 3 should be re-measured or explain the reason why this gallate has the highest ash content above 400°C. The seemingly highest ash content of propyl gallate can be the fluctuation of zero point.
Answer: The TGA measurement for propyl gallate has been re-measured.
Reviewer 1: (6) Chirality of lactic acid (L- or D-) in PLA and 3HB in P(3,4HB) should be specified. The average molecular weights and polydispersity index values of PLA and P(3,4HB), L-unit content of lactic acid, fraction and D-unit content of 3HB should be specified.
Yours sincerely,
PhD, DSc, Anna Masek
Associate Professor
Answer: We agree with the Reviewer. Data on the specificity of polymers are important, however, the manufacturers of the polymers used did not describe this data in the data sheets of the granulates. Based on the literature data we have determined that PLA containing 4.8 % D-lactide and average molecular weight (Mw) of 200 kDa. The P(3,4HB) containing 12 mol% 4-hydroxybutyrate and the average Mw was approximately 520 kDa. Data has been added to the manuscript.
Reviewer 1: (7) The authors say "the gallate was clearly observed on the SEM image in PLA matrix", which cannot be recognized. The authors are recommended to show the place of gallate with arrows or dotted circles.
Answer: Thank you for your important comment. We agree with the Reviewer. Marking the gallates in the SEM picture will improve its readability.
Reviewer 1: (8) The initial values of samples before ageing (at 0 h) is lacking in Figure 10.
Answer: Figures 10 and 11 refer to the change in color of the samples after aging. The color change coefficient dE*ab can be only determined for samples after aging. During spectrophotometric determination, samples before aging (0h) are reference samples for which the dE*ab factor is measured. For this reason, figures 10 and 11 cannot show the values of the samples before aging (at 0h).
Reviewer 2 Report
This manuscript introduces and analyzed the gallates as stabilizer for controlling the life time and degradation and as indicators within biodegradable polymer by several ways. It shows the possibility of petrochemical packaging materials environmentally. The investigation is really interesting and impressive. Moreover, this manuscript is well organized with experimental data and analysis methods. Therefore, I have no major problems with this work. Accordingly, I recommend this manuscript for publication in polymers after only minor revision. The following issues should:
1) It needs to compare the antioxidant properties between with gallates and without gallates in PLA and PHA system in Figure 2.
2) The WAXD spectrum for PHA samples showed the sharp diffraction peaks at 13.0 ˚, and 16.3 and 18.2 ˚, which corresponded to a crystalline nature polymer in Figure 6. However, there is a glass transition temperature (Tg) of PHA sample on the Table 2 although Tg concerns only amorphous polymers.
3) The mechanical properties were changed after thermo-oxidation and UV aging in Figure 7 and 8 and it was suggested that the reason was the different crystallinity states in the manuscript. However, it needs to present some evidence for explaining the change of the crystallinity states.
Author Response
Institute of Polymer and Dye Technology
Technical University of Lodz
90-924 Lodz, ul Stefanowskiego 12/16, Poland
Tel.: +48 42 631 32 23, Fax: +48 42 636 25 43
March 09, 2020
Professor
Receiving Editor
Polymers (MDPI)
Dear Professor,
We are resubmitting our revised paper entitled “Biodegradable polyester materials containing gallates” by Anna Masek and Malgorzata Latos-Brozio with a request to reconsider it for publication in " Polymers".
We have carefully considered the reviewers' comments. The manuscript was revised exactly according to these comments. The list of responses to the reviewer’s comments and corrections made in the manuscript is attached.
The manuscript has not been previously published, is not currently submitted for review to any other journal, and will not be submitted elsewhere before a decision is made by this journal.
For correspondence please use the following information:
corresponding author: Anna Masek
Institute of Polymer and Dye Technology
Technical University of Lodz
90-924 Lodz, ul Stefanowskiego 12/16, Poland
Tel.: +48 42 631 32 13
Fax: +48 42 636 25 43
e-mail: anna.masek@p.lodz.pl
Reviewer 2
Reviewer 2: This manuscript introduces and analyzed the gallates as stabilizer for controlling the life time and degradation and as indicators within biodegradable polymer by several ways. It shows the possibility of petrochemical packaging materials environmentally. The investigation is really interesting and impressive. Moreover, this manuscript is well organized with experimental data and analysis methods. Therefore, I have no major problems with this work. Accordingly, I recommend this manuscript for publication in polymers after only minor revision. The following issues should:
1) It needs to compare the antioxidant properties between with gallates and without gallates in PLA and PHA system in Figure 2.
Answer: We thank the Reviewer for comment. The determination of antioxidant properties, shown in Figure 2, concerned the determination only of gallate properties. The methodology for ABTS, DPPH, FRAP and CUPRAC determinations is not applicable and is not adapted to determine the antioxidant properties of polymeric materials. The analyzed substances require dissolution in the following solvents: water, ethanol or methanol. Dissolution of the polymers in this solvent is not possible, therefore the methods can not be used to determine the activity of polymeric materials.
Reviewer 2: 2) The WAXD spectrum for PHA samples showed the sharp diffraction peaks at 13.0˚, and 16.3 and 18.2˚, which corresponded to a crystalline nature polymer in Figure 6. However, there is a glass transition temperature (Tg) of PHA sample on the Table 2 although Tg concerns only amorphous polymers.
Answer: Crystalline polymers contain areas with ordered packing of macromolecules and amorphous areas, amorphous (supercooled liquid): according to the principle of thermodynamics, they have a two-phase structure. Crystalline polymers in which the crystalline phase is low should be called semi-crystalline polymers. In crystalline polymers in which there is no elastic region, the glass transition temperature Tg is equal to the melting point Tp.
PHA is a thermoplastic polymer. The WAXD results indicate the semicrystalline nature of the polymer, not a high degree of crystallinity, therefore PHA samples have a glass transition temperature.
Reviewer 2: 3) The mechanical properties were changed after thermo-oxidation and UV aging in Figure 7 and 8 and it was suggested that the reason was the different crystallinity states in the manuscript. However, it needs to present some evidence for explaining the change of the crystallinity states.
Answer: Thank you for the valuable comment. We agree with the Reviewer. The best method to show changes in the crystallinity of samples after aging would be the WAXD method. Unfortunately, we do not have this device in our institute and we order tests in other research institutions. Due to the short time (7 days) to improve the manuscript, we are not able to complete this research. We apologize for the lack of this research. We agree with the Reviewer that they are important and would increase the value of the manuscript. Thank you very much for the right suggestion. In the future, we will try to complete the analyzes with research suggested by the Reviewer.
Reviewer 3 Report
This manuscript studied the properties of gallates such as antioxidant capacity and thermal stability for PLA and PHA. Although this work has achieved some valuable results, there are some problems that should be clarified before further consideration.
- In page 6, line 227-235, why did the authors use T20, T50, and T90 to study the thermal stability? Basically, the thermal decomposition temperature at 1-5% weight loss can indicate the thermal stability.
- Can the authors comment on the different signal to noise ratio that is observed in WXRD patterns (Figure 6A and 6B)?
- There are no Figure 10D and 11D. Please carefully check these figures and captions.
- There are some misused words and spelling errors. For example, in abstract, “wide angle x-ray diraction…” should be “wide angle X-ray diffraction…”. It is better to use “morphology” to replace “topography” at page 7, line 243-257. As a result, this manuscript should be carefully checked by the authors.
Author Response
Institute of Polymer and Dye Technology
Technical University of Lodz
90-924 Lodz, ul Stefanowskiego 12/16, Poland
Tel.: +48 42 631 32 23, Fax: +48 42 636 25 43
March 09, 2020
Professor
Receiving Editor
Polymers (MDPI)
Dear Professor,
We are resubmitting our revised paper entitled “Biodegradable polyester materials containing gallates” by Anna Masek and Malgorzata Latos-Brozio with a request to reconsider it for publication in " Polymers".
We have carefully considered the reviewers' comments. The manuscript was revised exactly according to these comments. The list of responses to the reviewer’s comments and corrections made in the manuscript is attached.
The manuscript has not been previously published, is not currently submitted for review to any other journal, and will not be submitted elsewhere before a decision is made by this journal.
For correspondence please use the following information:
corresponding author: Anna Masek
Institute of Polymer and Dye Technology
Technical University of Lodz
90-924 Lodz, ul Stefanowskiego 12/16, Poland
Tel.: +48 42 631 32 13
Fax: +48 42 636 25 43
e-mail: anna.masek@p.lodz.pl
Yours sincerely,
PhD, DSc, Anna Masek
Associate Professor
Reviewer 3
Reviewer 3: This manuscript studied the properties of gallates such as antioxidant capacity and thermal stability for PLA and PHA. Although this work has achieved some valuable results, there are some problems that should be clarified before further consideration.
- In page 6, line 227-235, why did the authors use T20, T50, and T90 to study the thermal stability? Basically, the thermal decomposition temperature at 1-5% weight loss can indicate the thermal stability.
Answer: We agree with the Reviewer. The thermal decomposition temperature at 1-5% weight loss can indicate the thermal stability. However, in our considerations, we wanted to show the full thermal decomposition of the analyzed gallates, which is why we used T20, T50 and T90.
Reviewer 3: 2. Can the authors comment on the different signal to noise ratio that is observed in WXRD patterns (Figure 6A and 6B)?
Answer: Considerations regarding the results of WAXD measurements have been supplemented.
Reviewer 3: 3. There are no Figure 10D and 11D. Please carefully check these figures and captions.
Answer: Thank you for the comment. Figures have been corrected.
Reviewer 3: 4. There are some misused words and spelling errors. For example, in abstract, “wide angle x-ray diraction…” should be “wide angle X-ray diffraction…”. It is better to use “morphology” to replace “topography” at page 7, line 243-257. As a result, this manuscript should be carefully checked by the authors.
Answer: The errors indicated by the reviewer have been corrected. In addition, the entire manuscript has been carefully checked.
Round 2
Reviewer 1 Report
The following issues are remaining. The identical numbers as in the first review are given for the reviewer's comments.
(1) State that properties changes are partly caused by crystallization of PLA and P(3,4HB)in the manuscript.
(3) The authors' equation and maybe that in the literature can be applied only when no cold crystallization peak appears. As the author observed the cold crystallization peaks, they should use ΔHm-ΔHcc instead of ΔHm. If ΔHm-ΔHcc (or ΔHm+ΔHcc, depending on the sign of ΔHcc) values are minus as in the cases of some samples in the present study, the crystallinity values are zero. The crystallinity values in the present manuscript are those for the samples after cold crystallization not the initial samples before DSC measurements. See for example, Polymer 1995, 36, 2709.
(4) The authors explanation the causes for the unusual changes should be stated in the manuscript.
(6) Specify the configuration of 3HB component in the copolymer (L- or D-, or (R)- or (S)-) in the manuscript.
Author Response
nstitute of Polymer and Dye Technology
Technical University of Lodz
90-924 Lodz, ul Stefanowskiego 12/16, Poland
Tel.: +48 42 631 32 23, Fax: +48 42 636 25 43
March 13, 2020
Professor
Receiving Editor
Polymers (MDPI)
Dear Professor,
We are resubmitting our revised paper entitled “Biodegradable polyester materials containing gallates” by Anna Masek and Malgorzata Latos-Brozio with a request to reconsider it for publication in " Polymers".
We have carefully considered the reviewers' comments. The manuscript was revised exactly according to these comments. The list of responses to the reviewer’s comments and corrections made in the manuscript is attached.
The manuscript has not been previously published, is not currently submitted for review to any other journal, and will not be submitted elsewhere before a decision is made by this journal.
For correspondence please use the following information:
corresponding author: Anna Masek
Institute of Polymer and Dye Technology
Technical University of Lodz
90-924 Lodz, ul Stefanowskiego 12/16, Poland
Tel.: +48 42 631 32 13
Fax: +48 42 636 25 43
e-mail: anna.masek@p.lodz.pl
Reviewer 1
Reviewer 1: The following issues are remaining. The identical numbers as in the first review are given for the reviewer's comments.
(1) State that properties changes are partly caused by crystallization of PLA and P(3,4HB) in the manuscript.
Answer: Thank you for the comment. Sentence "The changes in mechanical properties were partly due to the crystallization of both polyesters" has been added to the manuscript.
Reviewer 1: (3) The authors' equation and maybe that in the literature can be applied only when no cold crystallization peak appears. As the author observed the cold crystallization peaks, they should use ΔHm-ΔHcc instead of ΔHm. If ΔHm-ΔHcc (or ΔHm+ΔHcc, depending on the sign of ΔHcc) values are minus as in the cases of some samples in the present study, the crystallinity values are zero. The crystallinity values in the present manuscript are those for the samples after cold crystallization not the initial samples before DSC measurements. See for example, Polymer 1995, 36, 2709.
Answer: The degree of crystallinity of the samples was once again calculated as suggested by the reviewer. Changes were introduced to the manuscript.
Reviewer 1: (4) The authors explanation the causes for the unusual changes should be stated in the manuscript.
Answer: We apologize for the oversight. An explanation “Antioxidant activity and the ability to reduce transition metal ions of all analyzed gallates change in the same way - as the concentration of gallate increases, the antioxidant capacity increases. Sometimes the difference between the oxidation activity of individual concentrations of the test compound is small, e.g. DPPH activity of lauryl gallate at 0.03 and 0.06mol/ml. However, these are not extraordinary changes. This behavior, during ABTS, DPPH, FRAP and CUPRAC determinations, is quite typical for substances of natural origin ” has been added to the manuscript.
Reviewer 1: (6) Specify the configuration of 3HB component in the copolymer (L- or D-, or (R)- or (S)-) in the manuscript.
Answer: The configuration of 3HB component in the copolymer is R-3HB. Sentence has been added in manuscript.